# Five Surfactin Isomers Produced during *Cheonggukjang* Fermentation by *Bacillus* *pumilus* HY1 and Their Properties

**DOI:** 10.3390/molecules26154478

**Published:** 2021-07-24

**Authors:** Su-Young Hong, Dong-Hee Lee, Jin-Hwan Lee, Md. Azizul Haque, Kye-Man Cho

**Affiliations:** 1SBT Business Division, Food Science Research Institute, Kolmar BNH Inc., Sejong 30003, Korea; syhong@kolmarbnh.co.kr; 2Industry Academy Cooperation Foundation, Andong National University, Andong 36729, Korea; dhlee@andong.ac.kr; 3Department of Life Resources Industry, Dong-A University, Busan 49315, Korea; schem72@dau.ac.kr; 4Department of Biochemistry & Molecular Biology, Hajee Mohammad Danesh Science & Technology University, Dinajpur 5200, Bangladesh; helalbmb2016@hstu.ac.bd; 5Department of Food Science, Gyeongsang National University, Jinju 52725, Korea

**Keywords:** *Bacillus pumilus* HY1, solid-state fermentation, *cheonggukjang*, surfactin, mass spectrum, cytotoxic effect

## Abstract

The cyclic lipopeptide produced from *Bacillus pumilus* strain HY1 was isolated from Korean soybean sauce *cheonggukjang*. The chemical structures of the surfactin isomers were analyzed using matrix-assisted laser desorption ionization time-of-flight mass spectrometry (MALDI-TOF MS) and electrospray ionization tandem mass spectrometry (ESI-MS/MS). The five potential surfactin isoforms were detected with protonated masses of *m/z* 994.7, 1008.7, 1022.7, 1036.7, and 1050.7 and different structures in combination with Na^+^, K^+^, and Ca^2+^ ions. ESI-MS/MS analysis revealed that the isolated surfactin possessed the precise amino acid sequence LLVDLL and hydroxyl fatty acids with 12 to 16 carbons. The surfactin content during c*heonggukjang* fermentation increased from 0.3 to 51.2 mg/kg over 60 h of fermentation. The mixture of five surfactin isoforms of *cheonggukjang* inhibited the growth of two cancer cell lines. The growth of both MCF-7 and Caco-2 cells was strongly inhibited with 100 μg/μL of surfactin. This study is the first-time report of five surfactin isomers of *Bacillus pumilus* strain HY1 during Korean soybean sauce *cheonggukjang* fermentation, which has cytotoxic properties.

## 1. Introduction

Surfactins are cyclic lipopeptide (CLP) biosurfactants produced by several *Bacillus* strains including *B. subtilis*, *B. amyloliquefaciens*, *B. pumilus*, *B. licheniformis*, and *B. mojavensis* [1,2,3,4,5]. Surfactin is produced non-ribosomally by multienzyme complexes called nonribosomal peptide synthetases [6]. The small microbial peptides produced via non-ribosomal pathways had demonstrated potent biological activities and growing economic value in uses such as drugs, e.g., antibiotics [7], cytotoxic benzolactones [8], alkaloids [9], or as food additives [10]. Surfactin is composed of one β-hydroxy fatty acid molecule with a long fatty acid moiety, linked to a seven amino acid peptide to form a lactone ring [11]. It exhibits diverse biological activities, including antitumoral [4,12,13], antiviral [14], antibacterial [15], antimycoplasmal [16], hemolytic [17], and fibrinolytic activities [18]. It also inhibits fibrin clot formation [19], phosphodiesterase activity [20], cyclic adenosine monophosphate (cAMP) signaling [20], and platelet and spleen cytosolic phospholipase A2 (PLA2) activity [21].

Fermented soybean is a good source of hydrolyzed peptides, protein, lipids, and many Koreans consume soybeans for their health benefits, which includes reducing arterial stiffness. There are several types of fermented soybean foods, including *meju* (soybean cake), *doenjang* (soybean paste), *kanjang* (soybean sauce), and *cheonggukjang* (soybean cook). Meanwhile, several reports have shown the antioxidant, antimicrobial, antigenotoxic, blood pressure lowering, and anti-diabetic activities of *cheonggukjang* [22,23]. In a previous study, we reported the production of surfactin from a potential probiotic, *Bacillus subtilis* CSY191, during the fermentation of *cheonggukjang*, and the purified surfactin exhibited anticancer activity against human breast cancer MCF-7 cells [4]. The identification and separation of the various beneficial metabolites in *chenoggukjang*, e.g., surfactin and micropeptide, enables a demonstration of their health benefits and promotes the development of distinct, functional products for the food industry.

In the present study, surfactin was purified from strain *B. pumilus* HY1, a bacterial strain commonly detected in traditional Korean fermented soy sauce (*kanjang*). The surfactin was fractionated through TLC and RP-HPLC and characterized using mass spectrometry. The concentration of surfactin during the fermentation of *cheonggukjang* by *B. pumilus* HY1 was determined. Finally, the antiproliferative activity of the surfactin extraction of *cheonggukjang* against the cancer cell lines MCF-7 (human breast cancer line) and Caco-2 (human intestinal cancer line) was studied.

## 2. Results

### 2.1. Isolation of Surfactin from B. pumilus HY1

The production spectrum of surfactin by the strain *Bacillus pumilus* HY1 was recorded and represented in the Figure 1. In fact, the negative control *Escherichia coli* DH5α lacks surfactin-producing genes, and did not shown any halo zone in the agar plate.

The surfactin was produced during the fermentation of high producer *B. pumilus* HY1 in No. 3 medium. Acid-precipitated and methanol-extracted bacterial cells were concentrated and chromatographed. The TLC of the obtained CLP material was performed on silica gel 60, using solvent (chloroform/methanol/water = 65:25:4, *v/v/v*) as the mobile phase, which showed a broad spot with a *Rf* value of 0.5 to 0.55. Analytical gel filtration of the entire surfactin mixture was performed using surfactin from *B. subtilis* (Sigma-Aldrich, Inc.) as a size marker, yielding a molecular mass of 1036. Reverse phase HPLC was used to analyze surfactin, followed by purification at 214 nm (Figure 2).

### 2.2. Mass Spectrum of Surfactin by B. pumilus HY1

The purified surfactin was analyzed and identified using MALDI-TOF MS, ICP MS, and ESI MS/MS. The MALDI-TOF MS displayed [M + H]^+^ peaks at *m/z* 994.7, 1008.7, 1022.7, 1036.7, and 1050.7, and these peaks were separated by *m/z* 14. The MALDI-TOF MS displayed the groups of peaks at *m/z* 1016.58, 1030.58, 1044.61, 1058.62, and 1072.62, which could reflect the isoform ensembles of surfactin (Figure 3). The surfactin contained a mixture of structural analogs with a mass difference of 14 Da. Most of the peaks could reflect essentially pure surfactin isoforms using MALDI-TOF MS to detect the distribution of the molecular ions in these fractions and the assignment of the ions to different surfactin species. Based on the mass spectrometric data, the surfactin isoforms were eluted according to their hydrophobicities. For the large peak appearing in fractions in front of the five main peaks, the main parent ions detected at *m/z* 1016.5811, 1030.5833, 1044.6104, 1058.6250, and 1072.6289 were attributed to a small amount of the sodium adduct of a valine-7 surfactin. The low mass peaks at *m/z* 1032.5869, 1046.5831, 1060.6001, 1074.6023, and 1088.6222 represented the potassium adduct of these species. The other mass peaks at *m/z* 1033.5630, 1047.5841, 1061.6024, 1075.5997, and 1089.8287 represented the calcium adduct of these species (Figure 3). These peaks contained essentially pure C_12_ to C_16_ surfactin species, respectively. The purified surfactin was analyzed using ICP MS, and potassium and calcium ions were detected (Table 1). These results were consistent with those of the MALDI-TOF MS analysis, suggesting the presence of ions. The four amino acids Asp, Glu, Val, and Leu at a ratio 1:1:1:4 were detected in the purified HPLC fraction of surfactin after hydrolysis and derivatization using Marfey’s reagent. The amino acid composition corresponded to that of surfactin. For the detailed analysis, the peptide sequence of surfactin was deduced after interpreting the ESI-MS/MS spectrum of the precursor ions *m/z* 994.7081, 1008.7334, 1022.7391, 1036.7528, and 1050.7487, assuming the preferential cleavage of the ring opening in the lactone bond in the collision chamber (Figure 4).

### 2.3. Changes in Viable Cell Number of B. pumilus HY1 and Surfactin Concentration during Cheonggukjang Fermentation

Changes in the bacterial viable cell numbers and surfactin concentration during the fermentation of *cheonggukjang* are shown in Figure 5. As a result of HPLC analysis in Figure 5a, the surfactin peak was detected at approximately 8.17 min. The level of viable cells in fermented *cheonggukjang* ranged from 3.0 log CFU/mL (0 h) to 11.7 log CFU/mL (60 h). Correspondingly, the concentration of surfactin in *cheonggukjang* fermentation increased from 0.3 mg/kg in the initial stage to 51.2 mg/kg after 60 h. The surfactin concentration peaked at 48.4 mg/kg at 48 h and increased slightly at the end of fermentation (Figure 5b).

### 2.4. Effect of MSIC on the Growth of Cancer Cells

We used the well-characterized MTT assay to assess the metabolic activity of cells, to determine whether surfactin could inhibit the growth of MCF-7 and Caco-2 cells, two cancer cell lines. MCF-7 and Caco-2 cells were incubated with MSIC at concentrations ranging from 0 to 120 μg/μL of surfactin, as shown in Figure 6. The MSIC inhibited the growth of cancer cells in a concentration-dependent manner. Morphological changes provide the most direct criteria for recognizing the apoptotic process. As shown in Figure 7, membrane blebbing and DNA fragmentation were observed 24 h after exposure to MSIC (100 μg/μL of surfactin). A protrusion-like shape began to form a branch as the shape of the cell lengthened. Moreover, it resulted in very severe morphological deformation of cells such as membrane shrinking and the rounding up of cells (Figure 7). As shown in Figure 8, DNA laddering, typical of the cells in which apoptosis occurred, was observed in the cancer cells with surfactin treatment. This means that endonuclease was activated by the surfactin treatment and then chromosomal DNA fragmented.

## 3. Discussion

In the present study, we investigated the purification of surfactin from *B. pumilus* HY1, the CLP (such as iturin and surfactin) producing strain isolated from *kanjang* [24]. The surfactins were fractionated using high resolution RP-HPLC and were determined by MADI-TOF MS and ESI-MS/MS. Moreover, the surfactin concentration was confirmed during *cheonggukjang* fermentation by *B. pumilus* HY1, and the cytotoxic effect of SECs against MCF-7 and Caco-2 cells was examined. Surfactin is a cyclic heptalipopeptide with the sequence Glu-Leu-Leu-Val-Asp-Leu, a terminal Leu linked to a β-hydroxy fatty acid residue, with amide (Glu) and lactone (Leu) bonds forming a cyclic structure. Three isoforms in natural surfactin with substitution of the L-Leu in position 7 of the main product by L-Val and L-Ile were characterized using 2D NMR spectroscopy in combination with chemical analysis [25]. The molecular network was distributed with known compounds from the Global Natural Products Social (GNPS) molecular networking standard library, where one molecular family (group of related precursor ions) contained the standard of the compound’s surfactin class [26]. The surfactin of *B. pumilus* HY1 possessed five potential isoforms with protonated masses of 994.7, 1008.7, 1022.7, 1036.7, and 1050.7 *m/z* and different structures in combination with Na^+^, K^+^, and Ca^2+^ ions (Figure 3). These results are consistent with data obtained from the analysis of the amino acid composition and showed that surfactin purified from *B. pumilus* HY1 was identical to surfactin produced from *B. subtilis* and *B. amyloliquefaciens* [4,27]. ESI-MS/MS analysis revealed that the structure of fatty acid was simulated, and the results indicated that surfactin was determined as a homologous series of hydroxy fatty acids with 12, 13, 14, 15, and 16 carbon atoms (Table 1). The biological activity of surfactins from *Bacillus* sp. depends both on the peptide ring and the nature of their lipid moieties. The hemolytic activity of surfactin was enhanced by the increase in the number of carbon atoms in the fatty acid side chains, and this enhanced activity likely reflects stronger interactions with biomembranes [28].

In previous studies, the biological activities of surfactin occurred via interactions with cellular membranes, and it may have a detergent-like effect on cell membranes [4,13]. The many research processes that have penetrated the hydrophobic residues of the peptide moiety permeate into the membrane interface, leading to membrane permeabilization but not surface effect [4,28]. At higher concentrations, this detergent-like activity could stabilize leaks due to the compound’s ability to generate multiple-structured polymers [28]. Although it is not yet clear whether a sub-optimal concentration (IC_50_ 100 μg/μL) of surfactin affects the cell morphology (Figure 7), surfactin-treated cells appeared to be leaky and/or lysed. Meta et al. [29]. proposed that surfactin may disturb a biochemical reaction(s) that occurs at a specific membrane site, perhaps via the long chain fatty acid, acting as a pseudosubstrate (similar to PIP3 or a farnesyl group) for either the PI3K/Akt or farnesyltransferase/Ras/ERK pathways, which promote apoptosis. The up-regulation of Bax, down-regulation of Bcl-xL, and activation of caspase-3 proteins could confirm the apoptosis of HL-60 cell lines using bioactive Cembranoids [30]. However, the expression level of these proteins was not studied in the present study, but the morphological changes, e.g., membrane blebbing and DNA fragmentation of the MSIC, which treated Caco-2 and MCF-7 cell lines, conclude that the potential apoptosis may occur. Alongside apoptosis, autophagy or senescence may be responsible for surfactin-induced growth arrests through the induction of either the p38-mediated MAPK pathway or ROS production [31,32]. In a previous study, Cao et al. [33] suggested that surfactin induces apoptosis in human breast cancer MCF-7 cells through a ROS/JNK-mediated mitochondrial/caspase pathway, and that the surfactin has notable anti-tumor effects on MCF-7 cells; however, there was no obvious cytotoxicity on normal cells.

Carbohydrate and lipoprotein extracts from edible mushrooms and several bacterial cells were reported for their growth inhibition towards malignant cancer cells [34]. A number of recent studies have focused on the anticancer activities of CLP. Kameda et al. [9] provided the first evidence that CLP has anticancer activity based on the fact that extracted CLP from *B. natto* KMD 2311 can induce apoptosis of cancer cells. 35. Wakamatsu et al. [34] reported that the exposure of B16 cells to lipoproteins resulted in chromatin condensation, DNA fragmentation, and sub-G1 arrest. Sudo et al. [35] examined lipoproteins for their ability to inhibit growth and induce differentiation of HL60 human promyelocytic leukemia cells, and Wakamatsu et al. [34] discovered that lipoproteins induce neuronal differentiation in PC12 cells and provided the groundwork for the use of microbial extracellular CLP as a novel reagent for the treatment of cancer cells. Wang et al. [36] reported that a new CLP purified from *B. subtilis* subsp. *natto* T-2 dose-dependently inhibited the growth of human leukemia K562 cells. Kim et al. [13] examined the effects of surfactin on the proliferation of LoVo cells (a human colon carcinoma cell line) and suggested that surfactin may have anticancer properties as a result of its ability to down-regulate the cell cycle and suppress cell survival. In addition, Cao et al. [37] reported that the purified surfactin from *Bacillus natto* TK-1 induced the time-dependent apoptosis of human breast cancer MCF-7 cells through cell cycle arrest at the G2/M phase. In sum, the above reports and the results of our work suggest that the surfactin of *B. pumilus* HY1 has potential value as a novel anti-tumor agent.

The content of surfactin increased from 0.3 mg/kg on day 0 to a final content of 51.2 mg/kg, corresponding to the cell concentration increases of 3.0–11.7 log CFU/g during *cheonggukjang* fermentation (Figure 5b). Several authors have studied the production of surfactin from solid-state fermentation [4,13,27,38]. This process is associated with the production of a higher concentration of surfactin in solid-state fermentation, which occurs in the manufacturing of soybean-fermented food [10]. In particular, Slvinski et al. [39] studied the production of surfactin through *B. pumilus* UFPEDA 448 in solid-state fermentation, using a medium based on okara with the addition of sugarcane bagasse as a bulking agent. This result shows that the concentration of surfactin is proportional to the concentration of cells because surfactin is produced from *Bacillus* sp.

The *cheonggukjang* is reported to have anticancer, blood pressure reduction, hypocholesterolemic, and fibrinolytic properties [4,22]. However, although these effects of *cheonggukjang* have been extensively studied, little is known about its potential antitumoral activities. In addition, we previously reported that the surfactin content, during *cheonggukjang* fermentation with *B. subtilis* CSY191, increased from 0.3 to 48.2 mg/kg over 60 h of fermentation, while the level of cytotoxic activity increased from 2.6- to 5.1-fold [4]. These findings will improve the quality of Korean traditional soybean fermented foods as one of the best functional foods because of their wide variety of antitumor, antimicrobial, antifungal, and antiviral activities.

## 4. Materials and Methods

### 4.1. Soybean, Media, Cells, and Regents

Soybean cultivar, namely Taekwang, was harvested and provided in 2013 from the National Institute of Crop Science of the Rural Development Administration in Korea. The tryptic soy (TS), number 3 (No. 3), and DMEM/F-12 media were purchased from Becton Dickinson Co. (Difco, Sparks, MD, USA). MCF-7 (human breast cancer cell line) and Caco-2 (human intestinal cancer cell line) were obtained from the Korean Cell Line Bank in Seoul, Korea. Sheep blood, surfactin standards (consisting of C_13_ and C_15_ β-amino acids), glycerol, triglycerol, α-cyano-4-hydroxycinnamic acid, trifluoroacetic acid, triethylamine, phenylisothiocyanate, 3-(4,5-dimethylthiazol-2-yl)-2,5-diphenyltetrazolium bromide (MTT), ethylenediaminetetraacetic acid (EDTA), dimethyl sulfoxide (DMSO), fetal bovine serum (FBS), penicillin, and streptomycin were obtained from the Sigma-Aldrich, Inc. (Merck KGaA, Darmstadt, Germany). The high-performance liquid chromatography (HPLC)-grade chloroform, methanol, water and acetonitrile were purchased from Fisher Scientific International, Inc. (Fairlawn, NJ, USA). Other regents and solvents used the analytical grade (Sigma-Aldrich, Inc., St. Louis, MO, USA).

### 4.2. Indentification of HY1 Strain, and Isolation and Purification of Surfactin

The cyclic lipopeptide producer strain HY1 was isolated from Korean soybean sauce (*cheonggukjang*). The *cheonggukjang* sample was serially diluted and grew on nutrient agar media to get a pure colony. The genomic DNA of the strain HY1 was extracted and used for amplification using Bacillus specific 16S rRNA primers: (478F 5′-TTCTACGGAGAGTTGATCC-3′; 479R, 5′-CACCTTCCGGTACGGCTACC-3′). The DNA sequencing and phylogenetic analysis of the HY1 strain was conducted as previously described [24]. Surfactin was isolated from strain HY1, according to Lee et al. [4] and Cho et al. [40]. The strain HY1 was cultured in 50 mL TS liquid medium at 30 °C for 48 h to prepare bacteria. The cells were grown in 10 L No. 3 medium (polypeptone 10 g, glucose 10 g, KH2PO4 1 g, and MgSO4∙7H_2_O 0.5 g per liter, pH 6.8) at 30 °C. After cultivation for 3 days, the supernatant was collected through centrifugation and adjusted to pH 2.0 using concentrated HCl. The precipitate was collected through centrifugation and extracted three times with methanol. The methanolic extracts were separated using thin layer chromatography on silica gel 60 plates (Merck KGaA, Darmstadt, Germany). Chloroform/methanol/water (65:25:5, *v/v/v*) was used as the developing solvent. The various spots were visualized by charring after spraying with concentrated sulfuric acid. To isolate the surfactin fraction, the corresponding spots were scratched from the thin-layer chromatography (TLC) plate and the silica gel material was extracted using methanol. The crude extract of surfactin was calculated at 3.25 gm/L of initial culture. For further purification, 1 mg of extract was subjected to recycling preparative HPLC (LC-908, Japan Analytical Industry, Co., Ltd., Tokyo Japan) and HPLC JAIGEL-1H column (C18, 250 × 94.6 mm, 5 µm, Phenomenex, CA, USA) and eluted with acetonitrile/water (1:1, *v/v*) as the mobile phase at a flow rate of 2.5 mL/min.

### 4.3. Mass Spectrometric Analysis of Surfactin Isoforms

The purified CLP was analyzed using matrix-assisted laser desorption ionization time-of-flight mass spectrometry (MALDI-TOF MS, Vg-Instruments, Manchester, UK) and an inductively coupled plasma mass spectrometer (ICP MS; Elan Drc II, PerkinElmer Inc., New York, NY, USA). Five micrograms of sample were dissolved in DMSO-glycerol and introduced on a copper probe tip using a mixture of glycerol and triglycerol as a matrix. A saturated solution of α-cyano-4-hydroxycinnamic acid in 70% acetonitrile/0.1% trifluoroacetic acid (1:1, *v/v*) was mixed with an equal volume of sample for the MALDI-TOF mass analysis. One microliter of the sample (2 to 3 *ρ*mol) was deposited onto a sample plate and air-dried. Ions were accelerated with a voltage of 20 kV. The positive-ion and reflector mode was applied.

### 4.4. Amino Acid and Mineral Analysis

The amino acid and mineral composition analysis of the purified CLP wines were determined according to methods previously described by Cho et al. [40]. The amino acid of the purified CLP was sequenced using electrospray ionization tandem mass spectrometry (ESI-MS/MS; Finnigan-MAT TSQ 700, San Jose, CA, USA). The sample was dissolved in 50% aqueous methanol containing 1% formic acid prior to injection into the mass spectrometer. ESI-MS/MS spectra were acquired after inducing collisions between precursor ions and nitrogen collision gas at acceleration voltages of 50 V.

### 4.5. Preparation and Fermentation of Cheonggukjang

Soybeans (1 kg) were washed and soaked with three times volume of tap water at 20 ± 2 °C for 12 h, followed by steaming for 30 min at 121 ± 1 °C. The steamed soybeans were incubated at 37 °C for 1 h to cool down the sample. The cooked soybeans were subsequently inoculated with 5% (*v/w*) HY1 (7.43 log CFU/mL), fermented for 60 h at 37 ± 2 °C and sampled at 0, 12, 24, 36, 48, and 60 h. The growth of strain HY1 during fermentation was determined as viable cell counts at each time point. One gram of the sample was mixed with 9 mL of 0.85% NaCl solution, and the dilutions were spread onto TS agar plates. Colonies were counted after incubation at 37 °C for 24 h [22].

### 4.6. Extraction and Analysis of Surfactin

The extraction and analysis of surfactin were performed according to Lee et al. [4] methods. Ten grams of ground *cheonggukjang* were extracted using 30 mL of methanol and adjusted to pH 2.0 with concentrated HCl by shaking (160 rpm) at 30 °C for 12 h. The extract was filtered through Whatman No. 2 filter paper and dried under a vacuum. The dried material was redissolved in 10 mL of 80% methanol and filtered through a 0.45-μm Millipore PVDF filter (Schleicher & Schuell, GmbH, Dassel, Germany). The filtrate was used for HPLC analysis to determine the surfactin concentration during the fermentation of *cheonggukjang*. The injection volume of the sample was 20 μL. The surfactin was analyzed through HPLC (Perkin–Elmer 200 series, Perkin–Elmer Corp., Norwalk, CT, USA) using an RP C18 column (4.6 × 250 mm, 5 μm, Waters Corp., Milford, MA, USA). The acetonitrile:water (1:1, *v/v*) was eluted at a flow rate of 1 mL/min at 40 °C. Surfactin was measured at 214 nm using a UV detector (Perkin–Elmer UV 200 series, Perkin–Elmer Corp., Norwalk, CT, USA). The concentration of surfactin was determined using a standard curve with standard solutions at 25, 50, 75, and 100 μg/mL.

### 4.7. MTT Assay

MCF-7 and Caco-2 cells were used as cancer cell lines for the MTT assay. The cells were cultured in DMEM/F-12 medium, supplemented with 10% FBS and 1% penicillin-streptomycin. The MTT assay was determined by the method of Lee et al. [4]. Briefly, the two cancer cells were dissociated with 0.05% trypsin-0.02% EDTA, and 180 μL of the cell suspension (1 × 10^4^ cells/mL) was seeded onto 48-well microtiter plates and treated with various concentrations of mixture of five surfactin isoforms of *cheonggukjang* (20 μL). After incubation for 24 h, 20 μL of MTT solution (2.5 mg/mL PBS) was added. The formazan dye was solubilized by adding 150 μL of DMSO to each well, followed by gentle shaking. The optical densities were read on an ELISA reader (680; Bio-Rad, Tokyo, Japan) at 540 nm. The analysis of DNA fragmentation was determined by an electrophoresis method.

## 5. Conclusions

We identified five potential isoforms of surfactin from *B. pumilus* HY1. Surfactin is a significant component of the stain HY1-fermented *cheonggukjang*, with surfactin reaching 51.2 mg/kg at the end of fermentation (60 h). Surfactin extraction of *cheonggukjang* had antiproliferative activity against two cancer cells lines. Further studies are needed to define the in vivo cytotoxic effects of surfactin and clarify its precise molecular mechanism(s) of action.

## Figures and Tables

**Figure 1 molecules-26-04478-f001:**
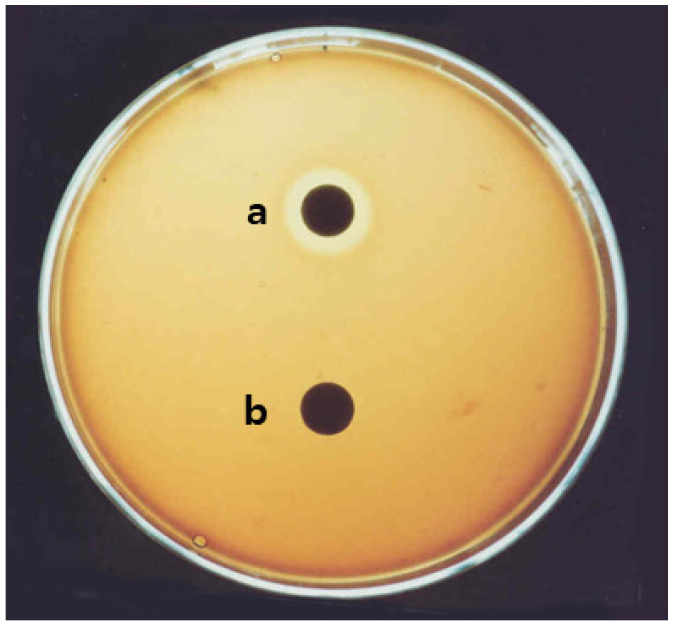
Surfactin production test of *Bacillus pumilus* HY1. The strain HY1 grown on defibrinogen sheep blood agar at 37 °C for 24 h: (**a**) *Bacillus pumilus* HY1; and (**b**) *Escherichia coli* DH5α. The *Escherichia coli* DH5α were used as negative control of surfactin production in agar plate.

**Figure 2 molecules-26-04478-f002:**
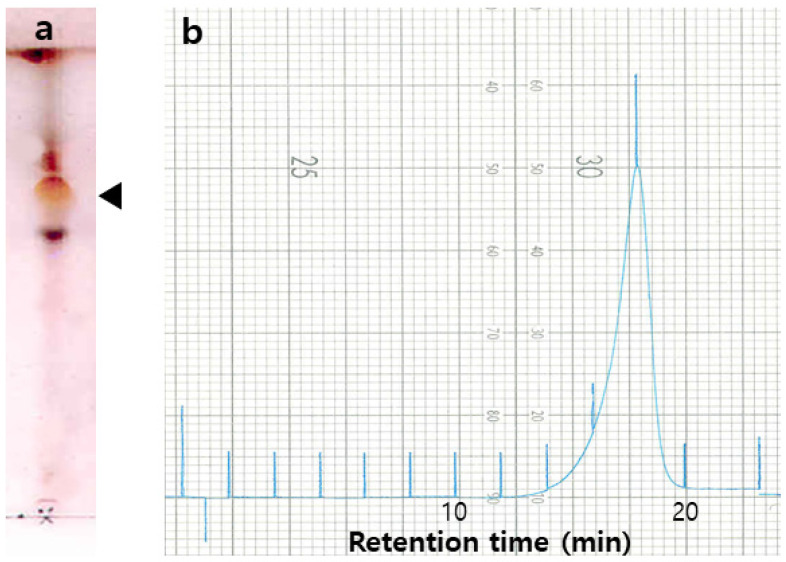
TLC and HPLC profiles of the surfactin produced by *Bacillus pumilus* HY1: (**a**) TLC profiles as follow: Developing solvent—chloroform:methanol:water = 65:25:4 (*v/v/v*) drops and visualization—10% H_2_SO_4_ in water; and (**b**) HPLC profiles as follow: Eluent solvent—acetonitrile:water = 1:1 (*v/v*), flow rate—2.5 mL/min, and absorbance—214 nm.

**Figure 3 molecules-26-04478-f003:**
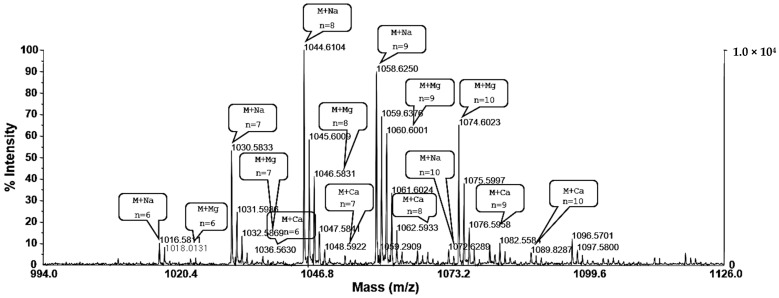
MALDI-TOF mass spectrum of the surfactin produced by *Bacillus pumilus* HY1: The lipopeptide appeared as a complex mixture of several isoforms. Sodium (Na), potassium (K), and calcium (Ca) ions were detected.

**Figure 4 molecules-26-04478-f004:**
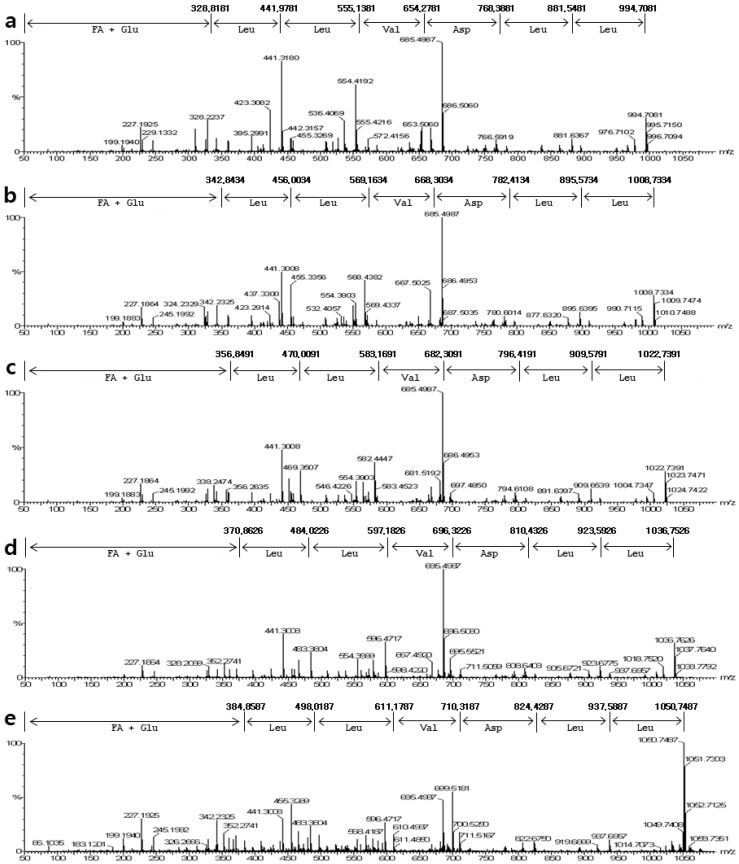
ESI tandem mass spectrometry fragmentation of surfactin isomers: Peptide sequences of five surfactin isomers were identified based on the MS/MS spectrum of the precursor ion *m/z* 994.7081, 1008.7334, 1022.7391, 1036.7528, and 1050.7487. (**a**) 994.7081 *m/z*; (**b**) 1008.7334 *m/z*; (**c**) 1022.7391 *m/z*; (**d**) 1036.7528 *m/z*; and (**e**) 1050.7487 *m/z*.

**Figure 5 molecules-26-04478-f005:**
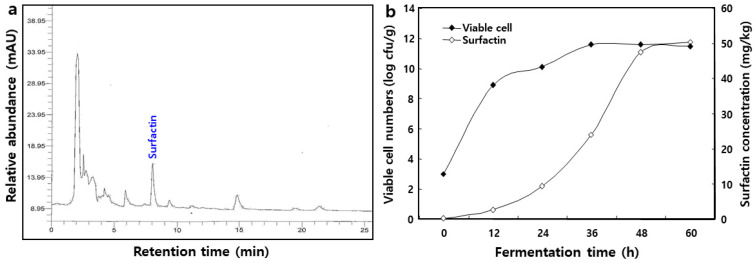
Typical HPLC chromatogram of the 95% methanol extracts in *cheonggukjang* (fermentation time: 60 h) and changes in the viable cell numbers and surfactin concentration during fermentation of *chenonggujang* by *Bacillus pumilus* HY1. (**a**) HPLC chromatogram; and (**b**) viable cell numbers and surfactin concentration.

**Figure 6 molecules-26-04478-f006:**
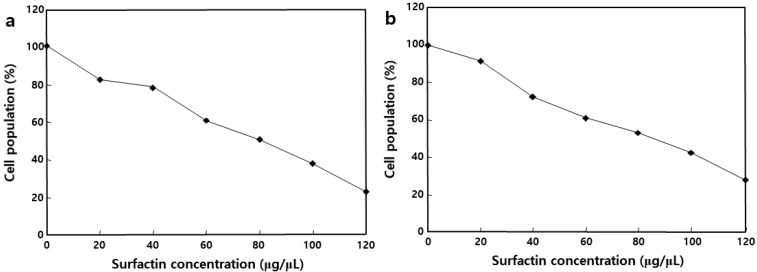
Cell population of human breast (MCF-7) and intestinal (Caco-2) cancer cell lines according to treatment of different surfactin concentrations: (**a**) mixture of five surfactin isoforms of *cheonggukjang*-treated MCF-7 cells; and (**b**) mixture of five surfactin isoforms of *cheonggukjang*-treated Caco-2 cells.

**Figure 7 molecules-26-04478-f007:**
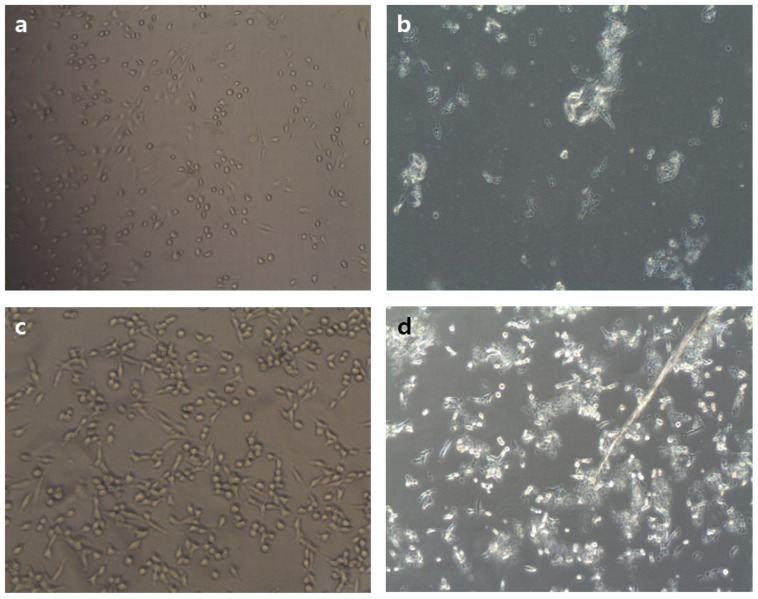
Cytotoxicity test of human breast (MCF-7) and intestinal (Caco-2) cancer cell lines by treatment of mixture of five surfactin isoforms of *cheonggukjang* (MSIC): (**a**) MCF-7 cells (control); (**b**) MSIC-treated MCF-7 cells (100 µg/µL); (**c**) Caco-2 cells (control); and (**d**) MSIC-treated Caco-2 cells (100 µg/µL).

**Figure 8 molecules-26-04478-f008:**
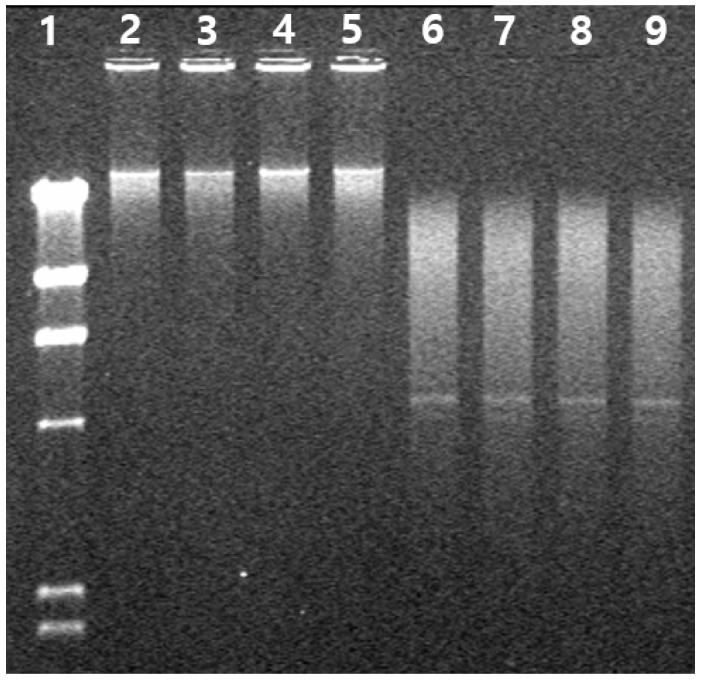
DNA fragmentation of human breast (MCF-7) and intestinal (Caco-2) cancer cell lines by treatment of mixture of five surfactin isoforms of *cheonggukjang* (MSIC): Line 1, size marker; line 2 and 3, MCF-7 cells (control); line 4 and 5, Caco-2 cells (control); line 6 and 7, MSIC-treated MCF-7 cells (100 µg/µL); and line 8 and 9, MSIC-treated Caco-2 cells (100 µg/µL).

**Table 1 molecules-26-04478-t001:** ICP mass spectrum of surfactin isomers from surfactin produced by *Bacillus pumilus* HY1.

Elements	Concentration (ppb)	Elements	Concentration (ppb)
Al	0.0	P	1121.5
B	133.6	Rb	0.0
Br	0.0	Re	0.0
Ca	1509.6	Ru	94.4
Cd	0.0	S	348,218.9
Cr	122,736.5	Sc	74,514.2
Cs	0.0	Se	1,112,974.3
Cu	0.0	Sn	115.1
Ge	194.9	Sr	0.0
Hg	26.9	Ta	2.9
I	1375.0	Ti	580.2
K	89.4	U	0.0
Li	122.4	V	0.0
Mg	5634.3	W	44.7
Mn	0.0	Y	538.1
Mo	0.0	Yb	0.0
Na	6446.2	Zn	2351.4
Os	0.0		

## Data Availability

Data reported in this study is contained within the article. The underlying raw data is available on request from the corresponding author.

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
