# Peer review of "Five Surfactin Isomers Produced during Cheonggukjang Fermentation by Bacillus pumilus HY1 and Their Properties"

_molecules, 2021, doi:10.3390/molecules26154478_

Round 1
Reviewer 1 Report
Dear Authors,
The work describe the presence of five surfactin isoforms in cheonggukjang associated Bacillus pumilus HY1. The structures of these surfactin analogs were proposed by MS/MS and Marfey's analyses. These surfactins showed cytotoxicity against MCF-7 and Caco-2 cell lines.
1) Page 1, introduction section, lines 40-42, "... NRPS are of great interest, ... many potent biological activities." This statement is a bit misleading that NRPS peptides are more interested due to their biological activities compared to RiPP peptide and polyketides. The sentence should be re-write and more references should be included, below are some references for NRPS peptides with cytotoxic and antibiotic properties.
J. Am. Chem. Soc. 2020, 142, 15, 7145–7152. DOI: 10.1021/jacs.0c01605
Angew. Chem. Int. Ed. Engl. 2020, 59, 21535-21540. Doi: 10.1002/anie.202009107.
Angew. Chem. Int. Ed. Engl. 2020, 59, 7766-7771. Doi: 10.1002/anie.201916007.
2) At 2.2 section, the method for identification of strain B. pumilus HY1 should be specified.
3) At 2.2 section, the crude extract yield and the volume of culture (liquid) or number of petri dishes (agar) should be specified.
4) At 2.2 section, HPLC column size should be specified. Also, the Marfey's analyses should be detailed out.
5) Figure 1, what is the justification of using E. coli DH5alpha? This is the competent cells that we used for cloning purpose.
6) Figure 2, the HPLC's UV detector was not in a good condition, the signals going up and down at each interval time. This situation can be improved by simply restart the UV detector, as we occasionally encountered this issue and solved by simply restart our old UV detector.
7) Page 6, line 203, Fig 4 is actually refer to Fig5?
8) Page 9, line 237, please refrain from using "anticancer", replace with cytotoxic.
9) Authors may consider to include the chemical structure of surfactin in this study, also may consider to use GNPS molecular network for the collected MS/MS spectra and searched against GNPS database for compound identification. An example show in below reference.
J. Nat. Prod. 2017, 80, 11, 2863–2873. DOI: 10.1021/acs.jnatprod.6b01185
10) Figure 6, cytotoxicity of surfactin extraction against MCF-7 and Caco-2 cell lines, is not clear as to what is surfactin extraction refer to. Does it mean crude extract, purified surfactin or mixture of five surfactin isoforms?
11) Page 7, lines 217-219, Figure 6 should be Figure 7. The sentence "As shown in Figure 6, apoptotic cells were observed at 24 h after exposure to SEC (100 μg/μL of surfactin)." proposed apoptotic was induced by the surfactin. But the method use is not enough to show that. Normally there are more experiment to do, and usually staining agent was used to show the morphological characteristic of chromatin condensation. Please refer to below reference.
Mar. Drugs 2018, 16(4), 99; DOI: 10.3390/md16040099
12) The novelty of this work should be emphasized in abstract, 100 ug/uL to inhibit MCF-7 and Caco-2 cell lines are not potent at all.
Author Response
The work describe the presence of five surfactin isoforms in cheonggukjang associated Bacillus pumilus HY1. The structures of these surfactin analogs were proposed by MS/MS and Marfey's analyses. These surfactins showed cytotoxicity against MCF-7 and Caco-2 cell lines.
→ Ans. Thank you for sir. We have edited according to previous reviewers and changes are highlighted in the red and blue (references) colors.
Q1. Page 1, introduction section, lines 40-42, "... NRPS are of great interest, ... many potent biological activities." This statement is a bit misleading that NRPS peptides are more interested due to their biological activities compared to RiPP peptide and polyketides. The sentence should be re-write and more references should be included, below are some references for NRPS peptides with cytotoxic and antibiotic properties.
→ Ans. Thank you very much sir for the valuable comment. After carefully reviewing the reference articles, we have revised the sentence as follows:
“The small microbial peptides produced via non-ribosomal pathways had demonstrated the potent biological activities and growing economic value such as drugs e.g. antibiotic [7], cytotoxic benzolactones [8], alkaloids [9] or food additives [10].” (lines 41-44)
References
- Li, H.; Gilchrist, C.L.M.; Phan, C.S; Lacey, H.J; Vuong, D.; Moggach, S.A.; Lacey, E.; Piggott, A.M.; Chooi, Y.H. Biosynthe-sis of a new benzazepine alkaloid nanangelenin a from Aspergillus nanangensis involves an unusual l-kynurenine-incorporating NRPS catalyzing regioselective lactamization. J. Am. Chem. Soc. 2020, 142, 7145-7152.
- Dose, B.; Ross, C.; Niehs, S.P.; Scherlach, K.; Bauer, J.P.; Hertweck, C. Food-poisoning bacteria employ a citrate synthase and a type II NRPS to synthesize bolaamphiphilic lipopeptide antibiotics. Angew. Chem. Int. Ed. 2020, 59, 21535-21540.
- Niehs, S.P.; Dose, B.; Richter, S.; Pidot, S.; Dahse, H.M.; Stinear, T.P.; Hertweck, C. Mining symbionts of a spider-transmitted fungus illuminates uncharted biosynthetic pathways to cytotoxic benzolactones. Angew. Chem. Int. Ed. 2020. 59, 7766-7771.
Q2. At 2.2 section, the method for identification of strain B. pumilus HY1 should be specified
→ Ans. Thank you for suggestion. According to your suggestion, we have revised those sentences as follows:
“2.2. Identification of B. pumilus strain HY1 and Isolation and purification of surfactin
The cyclic lipopeptide producer strain HY1 was isolated from Korean soybean sauce (cheonggukjang). The cheonggukjang sample was serially diluted and growth on nutrient agar media until gets a pure colony. The genomic DNA of the strain HY1 was extracted and used for amplification using Bacillus specific 16S rRNA primers: (478F 5´-TTCTACGGAGAGTTGATCC-3´; 479R, 5´-CACCTTCCGGTACGGCTACC-3´). The DNA sequencing and phylogenetic analysis of the HY1 strain was conducted as previously described [24].” (lines 86-93)
Q3. At 2.2 section, the crude extract yield and the volume of culture (liquid) or number of petri dishes (agar) should be specified.
→ Ans. Thank you for the comment. We have revised the sentences as follows:
“The crude extract of surfactin was calculated at 3.25 gm/L of initial culture.” (line 105)
Q4. At 2.2 section, HPLC column size should be specified. Also, the Marfey's analyses should be detailed out.
→ Ans. Thank you for the comment. We have revised the sentences as follows:
“The crude extract of surfactin was calculated at 3.25 g/mL of initial culture. For further purification, 1 mg of extract was subjected to HPLC (LC-908, JAIGEL-1H column, Japan Analytical Industry, Japan) column (C18, 250 × 9 4.6 mm, 5 µm, Phenomenex, CA, USA) and eluted with acetonitrile/water (1:1, v/v) as the mobile phase at a flow rate of 2.5 mL/min.” (lines 105-109)
Q5. Figure 1, what is the justification of using E. coli DH5alpha? This is the competent cells that we used for cloning purpose.
→ Ans. Thank you for comment.
“The Escherichia coli DH5α were used as negative control of surfactin production in agar plate. The figure 1 caption was edited accordingly.” (lines 171-172)
Q6. Figure 2, the HPLC's UV detector was not in a good condition, the signals going up and down at each interval time. This situation can be improved by simply restart the UV detector, as we occasionally encountered this issue and solved by simply restart our old UV detector.
→ Ans. Thank you for nice and comment and suggestion. We have added the HPLC chromatogram (Figure 5a) in Figure 5 and revised the sentences as follows:
“As a result of HPLC analysis in Figure 5A, the surfactin peak was detected at approximately 8.17 min.” (lines 221-222)
“Figure 5. Typical HPLC chromatogram of the 95% methanol extracts in cheonggukjang (fermentation time: 60 h) and changes in the viable cell numbers and surfactin concentration during fer-mentation of chenonggujang by Bacillus pumilus HY1. (a) HPLC chromatogram; and (b) viable cell numbers and surfactin concentration.” (lines 242-245)
Q7. Page 6, line 203, Fig 4 is actually refer to Fig5?
→ Ans. We have revised it as follows:
“Correspondingly, the concentration of surfactin in cheonggukjang fermentation increased from 0.3 mg/kg in the initial stage to 51.2 mg/kg after 60 h. The surfactin concentration peaked at 48.4 mg/kg at 48 h and increased slightly at the end of fermentation (Figure 5B).” (lines 224-226)
Q8. Page 9, line 237, please refrain from using "anticancer", replace with “cytotoxic”.
→ Ans. We have replaced anticancer, by “cytotoxic” in the text of our results.
Q9. Authors may consider to include the chemical structure of surfactin in this study, also may consider to use GNPS molecular network for the collected MS/MS spectra and searched against GNPS database for compound identification. An example show in below reference.
→ Ans. Thank you for suggestion. According to your suggestion, we have revised those sentences as follows:
“The molecular network was distributed with known compounds from the Global Natural Products Social (GNPS) molecular networking standard library, where one molecular family (group of related precursor ions) contained the standard of the compound's surfactin class [27].” (lines 267-270)
References
- Vallet, M.; Vanbellingen, Q.P.; Fu, T.; Caer, J.P.L.; Della-Negra, S.; Touboul, D.; Duncan, K.R.; Nay, B.; Brunelle, A.; Prado, S. An integrative approach to decipher the chemical antagonism between the competing endophytes Paraconiothyrium varialbile and Bacillus subtilis. J. Nat. Prod. 2017, 80, 2863-2873.
Q10. Figure 6, cytotoxicity of surfactin extraction against MCF-7 and Caco-2 cell lines, is not clear as to what is surfactin extraction refer to. Does it mean crude extract, purified surfactin or mixture of five surfactin isoforms?
→ Ans. Thank you for comment. We used mixture of five surfactin isoforms of cheonggukjang (MSIC) and edited the manuscript where necessary.
“Figure 6. Cell population of human breast (MCF-7) and intestinal (Caco-2) cancer cell lines according to treatment of different surfactin concentration. (a) mixture of five surfactin isoforms of cheonggukjang-treated MCF-7 cells; and (b) mixture of five surfactin isoforms of cheonggukjang-treated Caco-2 cells.” (lines 247-250)
Q11. Page 7, lines 217-219, Figure 6 should be Figure 7. The sentence "As shown in Figure 6, apoptotic cells were observed at 24 h after exposure to SEC (100 μg/μL of surfactin)." proposed apoptotic was induced by the surfactin. But the method use is not enough to show that. Normally there are more experiment to do, and usually staining agent was used to show the morphological characteristic of chromatin condensation. Please refer to below reference. Mar. Drugs 2018, 16(4), 99; DOI: 10.3390/md16040099
→ Ans. Thank you sir. We have understood our limitation and changed the sentence as follows:
“As shown in Figure 7, membrane blebbing and DNA fragmentation was observed at 24 h after exposure to MSIC (100 μg/μL of surfactin).”
“The up-regulation of Bax, down-regulation of Bcl-xL, and activation of caspase-3 proteins could confirm the apoptosis of HL-60 cell lines using bioactive Cembranoids [31]. The expression of these proteins were not studied in the present study, but the morphological changes e.g. membrane blebbing and DNA fragmentation of the MSIC treated Caco-2 and MCF-7 cell lines leads to conclude that the potential apoptosis may occur. Alongside with apoptosis, autophagy or senescence may responsible for surfactin unduced growth arrests through induction of either the p38 mediated MAPK pathway or ROS production [32, 33].” (lines 293-300)
References
- Kamada, T.; Kang, M.C.; Phan, C.S.; Zanil, I.I.; Jeon, Y.J.; Vairappan, C.S. Bioactive cembranoids from the soft coral genus Sinularia sp. in Borneo. Mar. Drugs. 2018, 16, 99.
- Yamakami, Y.; Kensuke, K.; Yonekura, R.; Kudo, I.; Fujii, M.; Ayusawa, D. Molecular basis for premature senescence induced by surfactants in normal human cells. Biosci. Biotechnol. Biochem. 2014, 78, 2022-2029.
- Zhao, H.; Yan, L.; Xu, X.; Jiang, C.; Shi, J.; Zhang, Y.; Liu, L.; Lei, S.; Shao, D.; Huang, Q. Potential of Bacillus subtilis lipopeptides in anti-cancer I: induction of apoptosis and paraptosis and inhibition of autophagy in K562 cells. AMB Expr. 2018, 8, 78.
Q12. The novelty of this work should be emphasized in abstract, 100 ug/uL to inhibit MCF-7 and Caco-2 cell lines are not potent at all.
→ Ans. We have edited the abstract as follows:
“Abstract: The cyclic lipopeptide produced from Bacillus pumilus strain HY1 was isolated from Korean soybean sauce cheonggukjang. The chemical structures of the surfactin isomers were analyzed using matrix-assisted laser desorption ionization time-of-flight mass spectrometry (MALDI-TOF MS) and electrospray ionization tandem mass spectrometry (ESI-MS/MS). The five potential surfactin isoforms were detected with protonated masses of m/z 994.7, 1,008.7, 1022.7, 1036.7, and 1,050.7 and different structures in combination with Na+, K+, and Ca2+ ions. ESI-MS/MS analysis revealed that the isolated surfactin possessed the precise amino acid sequence LLVDLL and hydroxyl fatty acids with 12 to 16 carbons. The surfactin content during cheonggukjang fermentation increased from 0.3 to 51.2 mg/kg over 60 h of fermentation. The mixture of five surfactin isoforms of cheonggukjang inhibited the growth of two cancer cell lines. The growth of both MCF-7 and Caco-2 cells was strongly inhibited with 100 μg/μL surfactin. This study first time report, five surfactin isomers of Bacillus pumilus strain HY1 during Korean soybean sauce cheonggukjang fermentation, which has cytotoxic properties.” (lines 15-27)

Reviewer 2 Report
Comments and Suggestions for Authors
The paper is interesing. It concerns the influence of the mixture of surfactin isoforms produced during cheonggukjang fermentation on the proliferation of two cancer cell lines: MCF - 7(human breast cancer line) and Caco-2 (human intestinal cancer line). Minor revision for the English language are requested (some phrases in the text are incomprehensible). Introduction is well written, but there is a lack of sentence linking two paragraphs. (Lines 48-49).
I wonder why the autors investigated the chemical structure of the surfactin isoforms obtained on the model medium (No. 3) instead containing cheonggukjang. It has been proved that the number and chemical structure of surfactin isoforms is determined by the production capacity of the strain, but also by the composition of the culture medium.
The research methods used in the work are described in detail (Materials and Methods section).
In the Result part, the text shoud be analyzed for compatibility with the numbering of the Figures.
There is a lack of information in manuscript regarding the analyzes presented in Figure 1.
Some suggestions:
Lines 153-154 The description in these lines does not apply to Figure 1 but to Figure 2.
Lines 165-169 Figure 3 instead Figure 2
Line 189 Figure 4 instead Figure 3
Line 203 Figure 5 instead Figure 4
Line 216 : remove surfactin; Figure 6 instead Figure 5
Line 218 Figure 7 instead Figure 6
Line 225-226 The description is incomprehensible.
Line 245 Figure 3 instead Figure 2
Line 248 from B. subtilis and B. amyloliquefaciens [?], probably, there is a lack of bibliographic date.
Line 287 Summarizing, the above reports and the results of our work suggest……
Author Response
Reviewer 2
The paper is interesing. It concerns the influence of the mixture of surfactin isoforms produced during cheonggukjang fermentation on the proliferation of two cancer cell lines: MCF - 7(human breast cancer line) and Caco-2 (human intestinal cancer line).
→ Ans. Thank you for sir. We have edited according to previous reviewers and changes are highlighted in the red, green and blue (references) colors.
Q1. Minor revision for the English language are requested (some phrases in the text are incomprehensible). Introduction is well written, but there is a lack of sentence linking two paragraphs. (Lines 48-49).
→ Ans. Thank you for comment. We have revised and reorganized the introduction (line 43 to…) as follows:
“Surfactins are cyclic lipopeptide (CLP) biosurfactants produced by several Bacillus strains including B. subtilis, B. amyloliquefaciens, B. pumilus, B. licheniformis and B. mojavensis [1-5]. Surfactin is produced non-ribosomally by multienzyme complexes called nonribosomal peptide synthetases [6]. The small microbial peptides produced via non-ribosomal pathways had demonstrated potent biological activities and growing economic value such as drugs e.g., antibiotic [7], cytotoxic benzolactones [8], alkaloids [9], or food additives [10]. Surfactin is composed of one β-hydroxy fatty acid molecule with a long fatty acid moiety, linked to a seven amino acid peptide to form a lactone ring [11]. It exhibits diverse biological activities, including antitumoral [4,12,13], antiviral [14], antibacterial [15], antimycoplasmal [16], hemolytic [17], and fibrinolytic activities [18]. It also inhibits fibrin clot formation [19], phosphodiesterase activity [20], cyclic adenosine monophosphate (cAMP) signaling [20], and platelet and spleen cytosolic phospholipase A2 (PLA2) activity [21].
Fermented soybean is a good source of hydrolyzed peptides, protein, lipids, and many Koreans consume soybeans for their health benefits, which include reducing arterial stiffness. There are several types of fermented soybean foods including meju (soybean cake), doenjang (soybean paste), kanjang (soybean sauce), and cheonggukjang (soybean cook). Meanwhile, several reports have shown the antioxidant, antimicrobial, antigenotoxic, blood pressure lowering and anti-diabetic activities of cheonggukjang [22, 23]. In a previous study, we reported the production of surfactin from a potential probiotic, Bacillus subtilis CSY191 during the fermentation of cheonggukjang, and the purified surfactin exhibited anticancer activity against human breast cancer MCF-7 cells [4]. The identification and separation of the various beneficial metabolites in chenoggukjang e.g. surfactin, micropeptide, enables a demonstration of their health benefits and promotes the development of distinct functional products for the food industry.” (lines 38-62)
References
- Li, H.; Gilchrist, C.L.M.; Phan, C.S; Lacey, H.J; Vuong, D.; Moggach, S.A.; Lacey, E.; Piggott, A.M.; Chooi, Y.H. Biosynthe-sis of a new benzazepine alkaloid nanangelenin a from Aspergillus nanangensis involves an unusual l-kynurenine-incorporating NRPS catalyzing regioselective lactamization. J. Am. Chem. Soc. 2020, 142, 7145-7152.
- Dose, B.; Ross, C.; Niehs, S.P.; Scherlach, K.; Bauer, J.P.; Hertweck, C. Food-poisoning bacteria employ a citrate synthase and a type II NRPS to synthesize bolaamphiphilic lipopeptide antibiotics. Angew. Chem. Int. Ed. 2020, 59, 21535-21540.
- Niehs, S.P.; Dose, B.; Richter, S.; Pidot, S.; Dahse, H.M.; Stinear, T.P.; Hertweck, C. Mining symbionts of a spider-transmitted fungus illuminates uncharted biosynthetic pathways to cytotoxic benzolactones. Angew. Chem. Int. Ed. 2020. 59, 7766-7771.
Q2. I wonder why the autors investigated the chemical structure of the surfactin isoforms obtained on the model medium (No. 3) instead containing cheonggukjang. It has been proved that the number and chemical structure of surfactin isoforms is determined by the production capacity of the strain, but also by the composition of the culture medium. The research methods used in the work are described in detail (Materials and Methods section).
→ Ans. Thank you for suggestion. First of all, it is difficult to separate and purify what is not separated from cheonggukjang because various phytochemicals exist in soybeans or cheonggukjang. For the purpose of isolating the surfactin produced by the bacteria, it was isolated by culturing in No 3 medium, which is a medium composition for producing antibiotics. We have revised and rewritten the materials and methods (Section 2.2.) as follows:
“2.2. Identification of B. pumilus strain HY1 and Isolation and purification of surfactin
The cyclic lipopeptide producer strain HY1 was isolated from Korean soybean sauce (cheonggukjang). The cheonggukjang sample was serially diluted and grew on nutrient agar media to get a pure colony. The genomic DNA of the strain HY1 was extracted and used for amplification using Bacillus specific 16S rRNA primers: (478F 5´-TTCTACGGAGAGTTGATCC-3´; 479R, 5´-CACCTTCCGGTACGGCTACC-3´). The DNA sequencing and phylogenetic analysis of the HY1 strain was conducted as previously described [24]. Surfactin was isolated from strain HY1 according to Lee et al. [4] and Cho et al. [25]. The strain HY1 was cultured in 50 mL TS liquid medium at 30 °C for 48 h to prepare bacteria. The cells were grown in 10 L No. 3 medium (polypeptone 10 g, glucose 10 g, KH2PO4 1 g, and MgSO4∙7H2O 0.5 g per liter, pH 6.8) at 30 °C.” (lines 86-96)
Q3. In the Result part, the text should be analyzed for compatibility with the numbering of the Figures.
→ Ans. Thank you for comment. We have confirmed the numbering of the Figures.
Q4. There is a lack of information in manuscript regarding the analyzes presented in Figure 1.
→ Ans. Thank you for comment. We have revised the 3.1 section as follows:
“The production spectrum of surfactin by the strain Bacillus pumilus HY1 was recorded and represented in the Figure 1. In fact, the negative control Escherichia coli DH5α lacks of surfactin producing genes did not shown any halo zone in agar plate.” (lines 166-168)
Q5. Some suggestions:
→ Ans. Thank you for suggestions. We have done it in the text.
Q5-1. Lines 153-154 The description in these lines does not apply to Figure 1 but to Figure 2.
→ Ans. We have reorganized the 3.1. section of results as follows:
“3.1. Isolation of surfactin from B. pumilus HY1
The production spectrum of surfactin by the strain Bacillus pumilus HY1 was recorded and represented in the Figure 1. In fact, the negative control Escherichia coli DH5α lacks of surfactin producing genes did not shown any halo zone in agar plate. (lines 165-168)
Figure 1. Surfactin production test of Bacillus pumilus HY1. The strain HY1 grown on defibrinogen sheep blood agar at 37 °C for 24 h. (a) Bacillus pumilus HY1; and (b) Escherichia coli DH5α. The Escherichia coli DH5α were used as negative control of surfactin production in agar plate. (lines 169-172)
The surfactin was produced during the fermentation of high producer B. pumilus HY1 in No. 3 medium. Acid-precipitated and methanol-extracted bacterial cells were concentrated and chromatographed. The TLC of the obtained CLP material was performed on silica gel 60 using solvent (chloroform/methanol/water = 65:25:4, v/v/v) as the mobile phase, which showed a broad spot with an Rf value of 0.5 to 0.55. Analytical gel filtration of the entire surfactin mixture was performed using surfactin from B. subtilis (Sigma-Aldrich, Inc.) as a size marker, yielding a molecular mass of 1,036. Reverse phase HPLC was used to analyze surfactin, followed by purification at 214 nm (Figure 2).” (lines 173-180)
Figure 2. TLC and HPLC profiles of the surfactin produced by Bacillus pumilus HY1. (a) TLC profiles as follow: Developing solvent - chloroform : methanol : water = 65 : 25 : 4 (v/v/v) drops and visualization - 10% H2SO4 in water; and (b) HPLC profiles as follow: Eluent solvent - acetonitrile : water = 1 : 1 (v/v), flow rate - 2.5 mL/min, and absorbance - 214 nm. (lines 181-185)
Q5-2. Lines 165-169 Figure 3 instead Figure 2
→ Ans. Thank you. We have replaced it.
Q5-3. Line 189 Figure 4 instead Figure 3
→ Ans. Thank you. We have replaced it.
Q5-4. Line 203 Figure 5 instead Figure 4
→ Ans. Thank you. We have replaced it.
Q5-5. Line 216: remove surfactin; Figure 6 instead Figure 5
→ Ans. Thank you. We have replaced it.
Q5-6. Line 218 Figure 7 instead Figure 6
→ Ans. Thank you. We have replaced it.
Q5-7. Line 225-226 The description is incomprehensible.
→ Ans. Thank you. We have revised and rewritten the Figure 6 and Figure 7 captions as follows:
Figure 6. Cell population of human breast (MCF-7) and intestinal (Caco-2) cancer cell lines according to treatment of different surfactin concentration. (a) mixture of five surfactin isoforms of cheonggukjang-treated MCF-7 cells; and (b) mixture of five surfactin isoforms of cheonggukjang-treated Caco-2 cells. (lines 246-250)
Figure 7. Cytotoxicity test of human breast (MCF-7) and intestinal (Caco-2) cancer cell lines by treatment of mixture of five surfactin isoforms of cheonggukjang (MSIC). (a) MCF-7 cells (control); (b) MSIC-treated MCF-7 cells (100 µg/µl); (c) Caco-2 cells (control); and (d) MSIC-treated Caco-2 cells (100 µg/µL). (lines 251-255)
Q5-8. Line 245 Figure 3 instead Figure 2
→ Ans. Thank you. We have replaced it.
Q5-9. Line 248 from B. subtilis and B. amyloliquefaciens [?], probably, there is a lack of bibliographic date.
→ Ans. Thank you for comment. We have edited the sentence by adding appropriate references as follows:
“These results are consistent with data obtained from the analysis of the amino acid composition and showed that surfactin purified from B. pumilus HY1 was identical to surfactin produced from B. subtilis and B. amyloliquefaciens [4, 28].” (lines 272-275)
Q5-10. Line 287 Summarizing, the above reports and the results of our work suggest……
→ Ans. Thank you for suggestion. We have edited the sentence as follows:
“In sum, the above reports and the results of our work suggest that the surfactin of B. pumilus HY1 has potential value as a novel anti-tumor agent.” (lines 322-324)

Reviewer 3 Report
The publication presented for review seems interesting and easy to receive. However, the correctness of the preparation, or rather the describing of the documentation, raises serious doubts. This needs to be improved.
Author Response
Reviewer 3
The publication presented for review seems interesting and easy to receive. However, the correctness of the preparation, or rather the describing of the documentation, raises serious doubts. This needs to be improved.
→ Ans. Thank you sir. We have edited according to previous reviewers and changes are highlighted in the red, green and purple colors.
Reviewer 4 Report
The manuscript concerns isolation of surfactin isomers and characterization of their biological effects on MCF-7 and Caco-2 cell lines. Various members of Bacillus genus are well known producers of surfactin. Surfactin was also demonstrated to exhibit cytotoxic and anticancer effect on cell lines. Therefore, the novelty of the presented study is limited.
1. Authors claim that extracted surfactin isomers exhibit anticancer activity. While these compounds were already shown to possess such activity, results presented in the manuscript do not support such statement. MTT assay which was used in the study enables detection of cytotoxic effects imposed by investigated compound. Based on MTT assay results and simple microscopy it is not possible to distinguish type of cell death observed in the experiments. Therefore assumption of apoptosis is not supported by presented results. To analyze apoptosis some appropriate assay should be used (annexin V, PARP cleavage, or at least formation of the apoptotic DNA ladder).
Moreover, presented results indicate, that extracted surfactins have cytotoxic effect on cell lines, not necessarily only cancer cell lines. Without control experiments which would include normal cells it is not possible to state that the actual anticancer effect is observed. Also comparison of the effective concentration parameters (EC50) in the case of normal and cancer cells can provide some indication regarding putative anticancer activity.
2. l.203 Authors refer to Figure 4 as to the figure presenting numbers of viable cells. Figure 4 presents results of mass spectrometry analysis, not determination of cell viability.
3. Figure 5. According to the vertical axis label and figure legend, the presented graph represents numbers of visible cells. I suppose that authors meant viable, not visible cells.
Author Response
Reviewer 4
The manuscript concerns isolation of surfactin isomers and characterization of their biological effects on MCF-7 and Caco-2 cell lines. Various members of Bacillus genus are well known producers of surfactin. Surfactin was also demonstrated to exhibit cytotoxic and anticancer effect on cell lines. Therefore, the novelty of the presented study is limited.
→ Ans. Thank you for sir. We have edited according to previous reviewers and changes are highlighted in the red, green, purple and blue (references) colors.
Q1-1. Authors claim that extracted surfactin isomers exhibit anticancer activity. While these compounds were already shown to possess such activity, results presented in the manuscript do not support such statement. MTT assay which was used in the study enables detection of cytotoxic effects imposed by investigated compound. Based on MTT assay results and simple microscopy it is not possible to distinguish type of cell death observed in the experiments. Therefore assumption of apoptosis is not supported by presented results. To analyze apoptosis some appropriate assay should be used (annexin V, PARP cleavage, or at least formation of the apoptotic DNA ladder).
→ Ans. Thank you for nice comment and advice. We will continue to reflect the points pointed out by the reviewers in future research. However, please understand that we cannot present it due to the lack of data at this paper. We have understood our limitation and changed the sentence as follows:
“As shown in Figure 7, membrane blebbing and DNA fragmentation was observed at 24 h after exposure to MSIC (100 μg/μL of surfactin).”
“The up-regulation of Bax, down-regulation of Bcl-xL, and activation of caspase-3 proteins could confirm the apoptosis of HL-60 cell lines using bioactive Cembranoids [31]. However, the expression level of these proteins was not studied in the present study, but the morphological changes e.g., membrane blebbing and DNA fragmentation of the MSIC treated Caco-2 and MCF-7 cell lines, conclude that the potential apoptosis may occur. Alongside apoptosis, autophagy or senescence may be responsible for surfactin-induced growth arrests through induction of either the p38 mediated MAPK pathway or ROS production [32, 33].” (lines 293-300)
References
- Kamada, T.; Kang, M.C.; Phan, C.S.; Zanil, I.I.; Jeon, Y.J.; Vairappan, C.S. Bioactive cembranoids from the soft coral genus Sinularia sp. in Borneo. Mar. Drugs. 2018, 16, 99.
- Yamakami, Y.; Kensuke, K.; Yonekura, R.; Kudo, I.; Fujii, M.; Ayusawa, D. Molecular basis for premature senescence induced by surfactants in normal human cells. Biosci. Biotechnol. Biochem. 2014, 78, 2022-2029.
- Zhao, H.; Yan, L.; Xu, X.; Jiang, C.; Shi, J.; Zhang, Y.; Liu, L.; Lei, S.; Shao, D.; Huang, Q. Potential of Bacillus subtilis lipopeptides in anti-cancer I: induction of apoptosis and paraptosis and inhibition of autophagy in K562 cells. AMB Expr. 2018, 8, 78.
Q1-2. Moreover, presented results indicate, that extracted surfactins have cytotoxic effect on cell lines, not necessarily only cancer cell lines. Without control experiments which would include normal cells it is not possible to state that the actual anticancer effect is observed. Also comparison of the effective concentration parameters (EC50) in the case of normal and cancer cells can provide some indication regarding putative anticancer activity.
→ Ans. Thank you for comment. We have replaced “anticancer effect” with “cytotoxic effect” in the text where necessary.
Q2. l.203 Authors refer to Figure 4 as to the figure presenting numbers of viable cells. Figure 4 presents results of mass spectrometry analysis, not determination of cell viability.
→ Ans. Thanks you for advice. We have changed the Figure numbering as follows:
“For the detailed analysis, the peptide sequence of surfactin was deduced after interpreting the ESI-MS/MS spectrum of the precursor ions m/z 994.7081, 1,008.7334, 1,022.7391, 1,036.7528, and 1,050.7487 assuming the preferential cleavage of the ring opening in the lactone bond in the collision chamber (Figure 4).” (lines 208-211)
Q3. Figure 5. According to the vertical axis label and figure legend, the presented graph represents numbers of visible cells. I suppose that authors meant viable, not visible cells.
→ Ans. Thank you for comment. We have replaced “visible cell numbers” with “viable cell numbers” in the Figure 5 and caption.
“Changes in the bacterial viable cell numbers and surfactin concentration during the fermentation of cheonggukjang are shown in Figure 5. As a result of HPLC analysis in Figure 5A, the surfactin peak was detected at approximately 8.17 min. The level of viable cells in fermented cheonggukjang range from 3.0 log CFU/mL (0 h) to 11.7 log CFU/mL (60 h).” (lines 220-226)
Figure 5. Typical HPLC chromatogram of the 95% methanol extracts in cheonggukjang (fermentation time: 60 h) and changes in the viable cell numbers and surfactin concentration during fermentation of chenonggujang by Bacillus pumilus HY1. (a) HPLC chromatogram; and (b) viable cell numbers and surfactin concentration. (lines 241-245)

Round 2
Reviewer 1 Report
Dear Authors,
All the major concerns are somewhat been addressed positively, I have no further suggestion.
Author Response
→ Ans. Thank you for nice comment and advice. We added the Figure 8 (DNA fragment image) and then have revised and reorganized the results section as follows:
“The analysis of DNA fragmentation was determined by an electrophoresis method.” (line 163)
“A protrusion-like shape began to form a branch as the shape of the cell lengthen. Also, it resulted in very severe morphological deformation of cells such as membrane shrinking and rounding up of cells (Figure 7). As shown in Figure 8, DNA laddering typical of the cells in which apoptosis occurred was observed by the cancer cells with surfactin treatment. This means that endonuclease was activated by surfactin treatment and then chromosomal DNA fragmented.” (lines 238-243)
Figure 8. DNA fragmentation of human breast (MCF-7) and intestinal (Caco-2) cancer cell lines by treatment of mixture of five surfactin isoforms of cheonggukjang (MSIC). Line 1, size marker; line 2 and 3, MCF-7 cells (control); line 4 and 5, Caco-2 cells (control); line 6 and 7, MSIC-treated MCF-7 cells (100 µg/µl); and line 8 and 9, MSIC-treated Caco-2 cells (100 µg/µL).

Reviewer 4 Report
The manuscript has been modified, nevertheless the introduced modifications did not clear issues raised in my first review.
Provided microscopic images do not support claim that apoptosis was visible. Moreover, with such a simple microscopy it is not possible to visualize DNA fragmentation, as authors wrote in the text. Also the microscopy as a method is not described in the manuscript.
In current form, the novelty of presented results is even more affected, since the fact of surfactin cytotoxicity is well known and described in the literature.
Author Response

(The authors gave the same response as above.)
